# Artificial Intelligence in Neuroradiology: A Review of Current Topics and Competition Challenges

**DOI:** 10.3390/diagnostics13162670

**Published:** 2023-08-14

**Authors:** Daniel T. Wagner, Luke Tilmans, Kevin Peng, Marilyn Niedermeier, Matt Rohl, Sean Ryan, Divya Yadav, Noah Takacs, Krystle Garcia-Fraley, Mensur Koso, Engin Dikici, Luciano M. Prevedello, Xuan V. Nguyen

**Affiliations:** 1Department of Radiology, The Ohio State University Wexner Medical Center, Columbus, OH 43210, USAluciano.prevedello@osumc.edu (L.M.P.); 2College of Medicine, The Ohio State University, Columbus, OH 43210, USA; 3College of Arts and Sciences, The Ohio State University, Columbus, OH 43210, USA

**Keywords:** artificial intelligence, neuroradiology, AI-based challenge competitions, machine learning, deep learning

## Abstract

There is an expanding body of literature that describes the application of deep learning and other machine learning and artificial intelligence methods with potential relevance to neuroradiology practice. In this article, we performed a literature review to identify recent developments on the topics of artificial intelligence in neuroradiology, with particular emphasis on large datasets and large-scale algorithm assessments, such as those used in imaging AI competition challenges. Numerous applications relevant to ischemic stroke, intracranial hemorrhage, brain tumors, demyelinating disease, and neurodegenerative/neurocognitive disorders were discussed. The potential applications of these methods to spinal fractures, scoliosis grading, head and neck oncology, and vascular imaging were also reviewed. The AI applications examined perform a variety of tasks, including localization, segmentation, longitudinal monitoring, diagnostic classification, and prognostication. While research on this topic is ongoing, several applications have been cleared for clinical use and have the potential to augment the accuracy or efficiency of neuroradiologists.

## 1. Introduction

Artificial intelligence (AI), including its subsets of machine learning (ML) and deep learning (DL), is currently one of the most heavily researched fields in radiology, including neuroradiology. The results of a recent PubMed database search for “artificial intelligence” and “neuroradiology” revealed that more papers were published in 2021 and 2022 than all years prior (Figure 1). In fact, neuroradiology accounts for nearly one-third of all AI-related papers in radiology [1]. Our analysis of the text in the titles of all PubMed entries from this query from 2017 onwards shows a relatively high proportion of deep learning articles, with brain imaging, particularly imaging for stroke, representing a common topic (Figure 2). With continued research and growing capital investment in the industry, AI has the potential to revolutionize the medical imaging industry by providing improved diagnostic accuracy, increased efficiency, reduced costs, and better patient outcomes. In contrast to the initial claims that AI would render radiologists obsolete, this outlook has recently been redefined into a new paradigm—that AI will augment the modern-day radiologist and facilitate higher-quality and more efficient patient care [2,3].

This paper will provide an overview of the current topics and advancements in artificial intelligence related to the field of neuroradiology, including ischemic stroke, intracranial hemorrhage, brain tumors, neurocognitive imaging, white matter lesions, spinal imaging, and head and neck imaging. It will primarily focus on the performance of new and innovative research tied to current or recent AI-based challenge competitions, with particular attention paid to tasks that are central to neuroradiologists and with an emphasis on the literature published in 2017 or later.

## 2. AI Challenge Competitions

AI challenge competitions are events organized around a specific task or problem. These competitions are designed to promote the research and development of AI by providing a platform for testing various algorithms and models against current best practices and other competing AI methodologies. They provide unique opportunities for researchers and vendors to benchmark their AI-based algorithms. Many current AI biomedical imaging challenges can be found on the website Grand Challenge (https://grand-challenge.org/, accessed on 15 May 2023). Radiology AI challenge competitions often make use of datasets of medical images of a given modality; for neuroradiology, the most commonly used radiologic modalities are computed tomography (CT) and magnetic resonance (MR) imaging, although radiography and positron emission tomography are occasionally included. In most cases, the datasets consist of images obtained via standard clinical protocols for processing and may also include metadata related to image acquisition parameters or other demographic or clinical data. Teams that submit candidate algorithms are typically permitted to perform any additional image processing steps they deem appropriate, such as masking or scaling. The design of these competitions varies depending on the scope and goals of each competition. Typically, candidate algorithms are expected to perform a specific computer vision task, such as lesion detection, segmentation, or classification, and public training datasets are made available to participants for this purpose; their performance is evaluated using a test set that is not made public during the competition.

A critical problem with AI in radiology is the lack of large, public, high-quality, and well-annotated datasets [4,5]. Research from single institutions or individuals is often based on algorithms that are optimized to relatively small home-grown datasets with limited generalizability or reproducibility. In some situations, large amounts of data are available but may not have been curated to ensure high data quality or adequate data standardization. Differing scanner types, imaging protocols, pre-processing techniques, and patient populations make validation and large-scale implementation difficult. A systematic review by Yu et al. demonstrated that a vast majority of published DL algorithms saw a drop in performance when deployed on external data [5]. AI-based challenges attempt to circumvent this problem and provide large, well-annotated datasets with which to test and validate various algorithms. For example, when planning a challenge related to tumor segmentation, the organizers would typically seek to acquire a large number of imaging exams of tumors, ideally from different institutions and geographic areas, and have experts annotate the pixels on the images that correspond to tumor. Similarly, for a lesion detection challenge, the organizers would obtain multiple imaging exams that were annotated by experts as positive or negative for the lesion of interest. While many of these datasets are still a work in progress, in our opinion, they are currently one of the most reliable methods with which to evaluate and compare various artificial intelligence methods and advancements.

## 3. Definitions

Artificial intelligence is an umbrella term for any machine or system that performs tasks typically associated with human intelligence. Machine learning (ML) is a subtype of AI and refers to a technique by which programs learn patterns and/or adapt based on experience, typically through the use of large training datasets. ML can be categorized into supervised, unsupervised, and semi-supervised versions. In supervised learning, the datasets are labeled by an expert to “train” the algorithm to produce a desired output, with each piece of data refining the algorithm further. This is one of the primary AI methods used in radiologic imaging. In unsupervised learning, the datasets are not labeled. Patterns within the dataset are inferred by the AI algorithm without prior information or guidance. As its name suggests, semi-supervised learning incorporates components from both supervised and unsupervised ML by utilizing both the labeled and unlabeled datasets during the AI algorithm’s training. Deep learning (DL) is a subtype of ML that uses multi-layer artificial neural networks, which are often greater than 20 layers deep [6]. A common DL method used in image classification and lesion detection tasks utilizes convolution neural networks (CNN), i.e., DL architectures designed for processing and analyzing images and organized like the structure of a human brain. AI approaches to segmentation often make use of variants of the U-Net architecture, which contains convolutional neural network components arranged such that an encoding arm extracts relevant features from an input image and a decoding arm subsequently converts the features into an image of similar size to the input image.

The evaluation and comparison of AI algorithms is inherently challenging. This is not only difficult due to variation in AI architectures, but also due to variation in test datasets. Fortunately, there are several performance metrics that allow for the standardization and comparison of AI algorithms, several of which are presented in this paper. Some of the metrics commonly used when discussing diagnostic accuracy include true positives (TP), false positives (FP), true negatives (TN), false negatives (FN), sensitivity [TP/(TP+FN)], specificity [TN/(FP+TN)], accuracy [(TP+TN)/(TP+TN+FP+FN)], precision [TP/(TP+FP)], Dice/F1 score [2TP/(2TP+FP+FN)], and the area under the receiver operating characteristic (ROC) curve (AUC). For image data, the Dice score, equivalent to the F1 score, is a common metric for evaluating model segmentation performance by evaluating the spatial overlap (pixelwise agreement) between a gold-standard segmentation and model-derived segmentation, with scores ranging from 0 (no overlap) to 1 (exact overlap). It represents the harmonic mean of precision and sensitivity and therefore is commonly used to capture both values within one metric.

## 4. Ischemic Stroke

Stroke is a leading cause of death and the number one cause of serious long-term disability in the United States, with ischemic stroke accounting for most stroke types [7]. Over the last decade, several large clinical trials have increased the time window for stroke intervention up to 24 h, significantly increasing the pool of potentially treatable stroke patients, with subsequent demand for the faster and more accurate interpretation of images [8,9,10].

AI advancements in ischemic stroke imaging have primarily focused on optimizing stroke workflows, the detection of large vessel occlusions (LVO), quantifying stroke scoring metrics, the segmentation and quantification of ischemic or at-risk tissue, and clinical stroke outcome prediction. This section will center on stroke segmentation, ischemic stroke detection, and the automation of stroke metrics. Current AI-based competitions and challenges related to stroke segmentation will also be discussed.

ML and DL have already made a significant impact on current clinical stroke imaging practices. Currently, there are at least 14 different commercially available AI-based software packages related to ischemic stroke imaging [11,12,13,14]. Most packages have received FDA (Food and Drug Administration) or EU (European Union) approval for use in tasks such as LVO detection, ASPECTS scoring, and perfusion analysis. These software packages primarily use ML and DL methods, including convolution neural networks (CNN), with many of them built upon established well-trained deep neural network architectures such as AlexNet, GoogleNET, ResNET, or DenseNet-121 [15,16].

### 4.1. Segmentation and Perfusion Imaging

A major goal of stroke imaging is to evaluate and accurately segment ischemic penumbra volume from core infarct volume. MRI/DWI is the gold standard for evaluating infarct core volumes. However, it is typically slow, expensive, and not always readily available. Additionally, manual segmentation of ischemic changes is cumbersome, time-consuming, and affected by inter-rater variability [17]. CT perfusion (CTP) is commonly performed for acute ischemic workup, often in tandem with computed tomography angiography (CTA), with relative cerebral blood flow (rCBF) and time-to-maximum (Tmax) among the most commonly used CTP metrics [9,10,18]. These metrics are derived from time–attenuation curves, with applied linear statistical models and threshold values used to develop perfusion maps and volumetric segmentation. Threshold cut-offs for core infarct and penumbra vary in the literature, but typical values include rCBF < 30% and Tmax > 6 s, respectively [18,19]. The prediction and segmentation of DWI core infarct volumes using CTP data is a heavily researched area in AI and remains an ongoing challenge [20,21].

Many major AI commercial software packages provide threshold-based perfusion maps and segmentation analysis via DL models. Over the last several years, multiple new architectures and methods have been developed for stroke segmentation for both CT and MR, many of which have been developed and tested during recent AI competitions and challenges.

The Ischemic Stroke Lesion Segmentation (ISLES) challenge (https://www.isles-challenge.org/, accessed on 15 May 2023) was created in 2015 as an open competition to develop and encourage the design of advanced tools for use in ischemic stroke analysis. The most recently completed challenge in 2018 evaluated 24 teams’ ability to segment infarcted tissue from CTP images, with corresponding DWI images being the referenced standard [22]. The dataset consisted of 103 cases, including 63 training cases and 40 test cases. A comparison was also made to a conventional threshold-based method (rCBF < 38%).

All the top-performing teams used various DL U-net architectures, with the winning team of Song et al. using a 3D multi-scarce U-shaped network [22,23,24,25]. Nearly every team outperformed the traditional threshold-based model using rCBF < 38% (mean Dice Score 0.34). However, the mean Dice score for the top-performing team was only 0.51, illustrating the persistent gap in performance and accuracy compared to the manual segmentation of DWI imaging. Soltanpour et al. recently proposed a MultiRes U-Net technique, in which contralateral CTP imaging and Tmax heatmaps were used as additional data with which to supplement the CTP input images. This study used the ISLES 2018 dataset and achieved a Dice score of 0.69, a significant improvement from the original competition [23]. A recent meta-analysis on the performance of ML segmentation in stroke found that top-performing algorithms used DL methods as opposed to conventional ML classifiers, with a pooled Dice score of 0.50 [26].

The 2022 ISLES challenge (https://isles22.grand-challenge.org/, accessed on 15 May 2023) is currently underway and has the goal of segmenting infarcted brain lesions using DWI, ADC, and FLAIR images. There have been over 90 entries from multiple countries, with the current highest DICE score of 0.78. An additional separate segmentation challenge using a larger standardized dataset of T1-weighted images known as ATLAS 2.0—Anatomical Tracing of Lesions After Stroke (https://atlas.grand-challenge.org/, accessed on 15 May 2023)—is also underway [27]. This challenge only has several entries, with the highest DICE score currently 0.61.

### 4.2. Large Vessel Occlusion (LVO) Detection and Stroke Scoring Metrics

Large-vessel occlusion (LVO) and a corresponding hyperdense artery sign are strong indicators of stroke on the CT angiogram (CTA) and non-contrast CT (NCCT), respectively. Many commercially available AI-based software packages provide LVO detection [11]. These are primarily confined to assessing only anterior circulation using single-phase CTA images and often use CNN-based algorithms [15]. Over the last few years, multiple studies have evaluated the performance of LVO detection for several available software packages with varying results. Sensitivities ranged from 73–98% and specificities ranged from 52–98% [28,29,30,31,32,33,34,35,36]. As a reference standard, in one study, neuroradiologists’ ability to detect anterior LVO using CTA and NCCT had sensitivities ranging from 75 to 88% and specificities ranging from 88 to 97% [37]. Those figures significantly increased when supplemented with CTP data. Recently, Stib et al. used multi-phase CTA and a CNN using a DenseNet-121 architecture, with sensitivities of up to 100% and specificities of 77% when using all three phases [38]. In contrast to other studies, posterior circulation and cervical ICA occlusions were included in the analysis. Just as it is for radiologists, AI-based algorithms have low sensitivity for detection of distal occlusions (i.e., M2 or M3) [31,33,35].

The Alberta Stroke Program Early CT Score (ASPECTS) is a quantitative 10-point scoring system used to assess patients who would benefit from endovascular therapy. The score is determined based on early ischemic changes visible on NCCT [39]. The scale assesses 10 regions of the MCA territory, subtracting (from a normal score of 10) one point for every affected area, and patients with a resultant score of greater than or equal to six are given priority for thrombectomy [8]. Unfortunately, an inherent problem with ASPECTS is poor inter-rater variability between radiologists [40]. Many commercially available packages provide ASPECTS scoring, several of which employ classical ML methods like random forests [14]. DL methods are also being explored, with encouraging results [41]. Several studies have demonstrated AI ASPECTS scoring performance to be equal to that of experienced neuroradiologists [42,43,44,45,46,47]. Albers et al. showed that RAPID ASPECTS was more accurate than experienced readers in identifying early ischemia when compared to the corresponding DWI results [48]. Unfortunately, AI software for LVO detection and ASPECTS scoring have difficulty interpreting images of patients with findings of chronic underlying abnormalities, such as remote infarcts, chronic white matter changes, and post-operative changes [11,49].

The various available automated tasks described in this section have potential to facilitate the objective characterization of ischemic stroke to guide emergent treatment. While the overall performance of these methods is encouraging, the role of the experienced radiologist remains vital to the final interpretation and diagnosis.

## 5. Intracranial Hemorrhage

In the acute clinical setting, intracranial hemorrhage (ICH) represents a potentially life-threatening situation that demands prompt and accurate detection with imaging. Failure to detect and diagnose ICH in a timely manner can result in a delay of treatment and thus higher morbidity and mortality, underscoring the importance of early and accurate detection [50,51,52]. Coupled with increasing imaging workloads and a shortage of qualified radiologists, the need to streamline and expedite reads is paramount. Current AI research on ICH is aimed at ICH detection, segmentation, and classification, the prediction of hemorrhagic expansion, and even workflow prioritization. This section will focus on recent AI advancements in the detection and segmentation of ICH.

### 5.1. Detection

Due to its high sensitivity and specificity, CT is the modality of choice for ICH detection. Many recent studies on ICH detection examined DL neural networks, and most studies evaluating specific algorithms were retrospective, often comparing the results to those obtained with experienced neuroradiologist(s). Overall, current software packages have performed well with ICH detection. For example, Heit et al. evaluated RAPID ICH detection of ICH compared to the results of 3 trained neuroradiologists, with sensitivities and specificities of 96 and 95%, respectively [53]. This study included all subtypes of hemorrhage, including subdural, epidural, subarachnoid, intraparenchymal, and intraventricular. False-negative cases were primarily associated with small-volume hemorrhaging (<1.5 mL). False positives were associated with the observation of calcifications and other hyperdense structures on CT. Colasurdo et al. used the Viz.ai software package to detect subdural hemorrhage (SDH). The package had a sensitivity and specificity of 91% and 96%, respectively, with sensitivities smaller for small chronic SDH [54]. Another study, by Seyam et al., used Aldoc Medical to detect subtypes of hemorrhage. A sensitivity of 87.2% was observed with a 97.8% NPV. The program yielded lower detection rates for subdural hemorrhage (69.2%) and subarachnoid hemorrhage (77.4%) [55]. Whereas most studies to date have used small datasets from a single institution, Matsoukas et al. in 2022 performed the first systemic review of ICH detection, summarizing and pooling the performance results of various AI algorithms over the last couple decades. This review included approximately 40 relevant studies and reported the sensitivity and specificity for ICH detection to be 92% and 94%, respectively [56].

Numerous software tools have been developed in the last decade, many with high-performance and expert-level accuracy in terms of the diagnosis of ICH. Implementing them into clinical practice could improve the quality of radiology workflow, effectively providing a “second” read or quality assurance to the reading radiologist. One major obstacle to robust performance, however, lies in single-center design and the small size of single-institution training and test datasets, limiting external validity and applicability to a larger-scale setting. One initiative to improve AI-based algorithms would be the creation of large-scale annotated datasets. AI challenge competitions provide this level playing field, allowing the nonbiased and validated performance comparison of algorithms’ performance.

The first AI ICH detection challenge took place in 2019 when the Radiological Society of North America (RSNA) collaborated with the American Society of Neuroradiology (ASNR) [57]. An 874,035-image brain hemorrhage CT dataset was pooled from historical imaging from Stanford University, Universidad Federal de Sao Paulo, and Thomas Jefferson University Hospital [58]. This dataset was annotated by 60 volunteer neuroradiologists, serving as the reference standard, with each annotator labeling each imaging set as one of the following: ICH (and its subtype), normal, or abnormal but no hemorrhage. Each participating team’s algorithm was subsequently evaluated and ranked based on its ability to detect and classify ICH. Using the 2019-RSNA batch-1 test set, the winning team had sensitivity and specificity scores of 0.950 and 0.944, respectively, for all subtypes of ICH [59]. Detection metrics were worst for SDH and best for IVH. The 2019 ICH challenge provided scale, the opportunity for comparisons, and external validity to ICH detection algorithms. It also offered full transparency into the details of the various algorithms.

### 5.2. Segmentation

In addition to detection, hemorrhagic volume and expansion are important predictors of outcomes and treatment responses [60]. Accurate measurements of ICH, IVH, and perihematomal edema (PHE) facilitate the prediction of morbidity and mortality [60]. A common method for ICH volume calculation is known as the ellipsoid approximation technique. This involves the product of 0.52 and manual measurements by the radiologist for the 3 orthogonal axes. One study found that this method leads to the overestimation of hemorrhage size by nearly 20%, particularly when dealing with large or irregularly shaped hematomas [61]. Other limitations include significant interrater variability, limiting reliability, and the temporal demands of manual ICH measurements.

In recent years, researchers have investigated the ability of ML models to mitigate the aforementioned shortcomings of manual ICH segmentation. Patel et al. 2019 found that a convolutional neural network was able to match radiologists’ estimation of ICH volume [61]. Islam et al. in 2019 developed a novel DL model known as ICHNet. This achieved a Dice score of 0.89, comparable to those obtained by radiologists [62]. Heit et al., using RAPID, demonstrated results that strongly correlated with the expert consensus (r = 0.983) [53]. DL models based on the U-Net architecture or derivatives have also been used in segmentation. Zhao et al. utilized a no-new-U-Net (nn-Unet) framework, a type of DL model without a predetermined internal organization. Instead, this type of model continuously adapts to its training data and is therefore better optimized for a wider variety of data cohorts. Zhao reported Dice scores of 0.92 for ICH, 0.79 for IVH, and 0.71 for PHE [60]. They suspected that lower scores for IVH might be related to limited test data. Kok et al. evaluated various types of algorithms used in segmentation using a large-scale multicenter database from the Tranexamic Acid for Hyperacute Primary Intracerebral Hemorrhage (TICH-2) trial [63]. The types of algorithms studied and compared were U-Net, nnUnet, BLAST-CT, and DeepLAb3+. All these DL algorithms represent well-qualified, state-of-the-art technology with excellent track records in medical AI. They concluded that U-Net-based networks achieved significantly better performances than others for ICH and intraventricular hemorrhage (IVH) segmentations (*p* < 0.05). The top-performing models in their study had median Dice scores of 0.92 for ICH, 0.66 for perihematomal edema (PHE), and 1.00 for IVH. Notably, a nnU net algorithm, named Focal, achieved the highest Dice score with IVH. Not unexpectedly, a worse performance was noted for the detection of smaller hemorrhages.

## 6. Brain Tumors

Primary central nervous system (CNS) tumors are relatively rare neoplasms, with potentially devastating outcomes. Intracranial metastatic disease is the most common CNS tumor overall, accounting for more than half of all brain malignancies. Gliomas and meningiomas account for two thirds of all primary CNS tumors. High-grade gliomas (glioblastomas) are an often-catastrophic malignancy, with persistently low survival rates despite our improved understanding of the disease [64,65]. On the other hand, brain metastasis survival rates and incidence rates are continuing to increase, which is likely related to a combination of improved therapeutic regimens and tumor detection [66,67,68]. Imaging is therefore continuing to play an ever-increasing and vital role in both the detection and monitoring of disease.

Advancements in AI for brain tumors are broad and include research into tumor detection (identifying and/or localizing a tumor within an image), segmentation (defining tumor boundaries), grading (determining aggressiveness of a tumor), prognostication (predicting future outcomes such as survival), and treatment response assessment (evaluation of pseudo-progression and post-treatment necrosis). Several research areas are related to an emerging field called radiogenomics (imaging genomics), which explores the correlations between imaging characteristics and genetic/mutational patterns, essentially providing a “virtual biopsy” based on imaging patterns and other clinical data [69]. Although radiogenomics holds great potential, this section will primarily focus on advancements in AI that are central to a neuroradiologist’s role, such as tumor detection, segmentation, and post-treatment disease evaluation.

### 6.1. Tumor Detection

Tumor detection and surveillance of disease is a common and important task for radiologists. Targeted therapies like stereotactic radiosurgery (SRS) have pushed radiologists to meticulously detect and track numerous CNS lesions, often adding significant time and effort for each study. In a time-constrained work environment, the potential to augment tumor detection via AI is appealing. Two recent meta-analyses by Ozkara et al. and Cho et al. demonstrated that classical ML and DL algorithms overall performed well in the detection of brain metastasis, with a pooled sensitivity of 89–90% [70,71]. As a reference, 7 board-certified radiologists and 5 resident radiologists in one study had a similar pooled sensitivity of 89% [72]. However, sensitivity was significantly worse with small lesions. This is discouraging as the detection of smaller lesions is likely to provide the most benefit for radiologists [73]. Most DL algorithms within this analysis used 3D U-net or Deep Medic architectures. While the sensitivities are encouraging, it is unlikely this technology could replace trained radiologists, but rather augment their work. In one multi-center study performed using a multi-scale CNN detection algorithm to assist radiologists, the sensitivity for radiologists to detect brain metastasis increased by 21%, with a decreased reading time of 40% [74]. This type of result is likely to be the most realistic benefit of AI in this field, namely, the development of a tool to improve the quality and efficiency of radiologists.

### 6.2. Tumor Segmentation

The segmentation of brain tumors involves accurately and reliably delineating brain tumors from normal tissue, other disease pathologies, and imaging artifacts. The segmentation of brain tumors is difficult due to the variable size and heterogenous appearance of CNS malignancies. Additionally, some tumors demonstrate poor contrast enhancement and can easily overlap in intensity with the surrounding healthy brain parenchyma. Tumor grading, disease prognosis, treatment options, and post-treatment monitoring all depend on accurate segmentation. To explore AI facilitation of these efforts, the Brain Tumor Segmentation (BraTS) challenges have been ongoing for 10 years, representing the crucial joint efforts of the Radiological Society of North America (RSNA), the American Society of Neuroradiology (ASNR), and the Medical Image Computing and Computer Assisted Interventions (MICCAI) society to characterize gliomas and to improve algorithms designed for brain glioma segmentation, tumor compartmentalization, and tumor molecular characterization. New dataset analytics, including the most recent iterations of the BraTS challenge, have expanded case pathology and segmentation analytics beyond just gliomas to include meningiomas, pediatric brain, tumors, brain metastases, schwannomas, and other brain tumors [75].

While there are several recognized brain tumor datasets, the BraTS datasets are some of the most widely used for training and testing AI-based algorithms that focus on tumor segmentation. These multi-institution datasets have been growing annually since 2012 and comprise hundreds of well-annotated images obtained using common MR sequences (T1W, T2W, T1c, and FLAIR) of both high- and low-grade tumors. The datasets also consist of pooled ground truths determined by a set of expert raters who segment various components including edema, necrotic tumor core, and enhancing tumor. Numerous studies and papers related to tumor segmentation have been published in relation to the BraTS dataset over the years using various AI methods, including DL and other ML algorithms, with performance improving over the years [76,77]. Initial algorithms using the 2012 dataset primarily used conventional ML methods, with Dice scores ranging from 0.14 to 0.70. Recently, DL models have dominated the landscape, with significant performance increases. One recent meta-analysis of 10 ML-based segmentation studies demonstrated Dice scores of 0.84 [78]. Another meta-analysis of DL algorithms demonstrated slightly improved performance, with a pooled Dice score of 0.89 [76]. A recent study using the BraTS datasets and a 3D U-Net DL method achieved a Dice score of 0.95 [79]. There are countless additional algorithms that have achieved results above 90%, illustrating the significant progress made in such a short period of time.

### 6.3. Post Treatment Evaluation

Radiomics studies are using ML to assess brain tumor responses to clinical treatment, differentiate progression and pseudoprogression, and predict the recurrence and infiltration of neoplastic disease. It can often be hard to delineate tumor response and post-chemotherapy and radiation changes upon subsequent imaging. Further complicating the picture are new immunotherapies that incite complex inflammatory responses and antiangiogenic agents like bevacizumab that can cause a reduction in enhancement by reducing microvascular growth, but without necessarily offering any improvement in the overall survival rate [4].

Pseudoprogression is characterized by increased extent of imaging abnormalities after treatment, including abnormal enhancement and T2/FLAIR signal changes, that typically resolve without intervention. Differentiating pseudoprogression from true tumor progression is vital to treatment success and clinical outcomes. AI methods and techniques, including radiomics, are currently being explored to differentiate true progression from pseudoprogression by incorporating features from advanced MRI techniques, including diffusion-weighted imaging (DWI), perfusion-weighted imaging (PWI), and MR spectroscopy [80]. For example, Sun et al. performed a retrospective study on 77 post-treatment GBM patients, 26 of whom had known pseudoprogression [81]. Clinical data, patient outcomes, and multiple features from T1-weighted imaging were used to develop a radiomics model in order to evaluate true progression from pseudoprogression in comparison to the results of 3 trained radiologists. The model demonstrated sensitivity and specificity of 78% and 61%, respectively. The 3 radiologists had sensitivities of 62–69% and specificities of 47–68%.

Differentiating infiltrating neoplasms from adjacent edema is difficult using conventional imaging approaches. However, ML has the capability to identify the margins of infiltrative tissue from normal tissue on pre-treatment MRI images. Identifying infiltrative margins is important in resection planning, biopsy site selection, and monitoring treatment response. Approaches that incorporate ML have been successful in generating spatial maps of infiltrated tissue, with approximately 90% cross-validated accuracy [82]. A fully automated CNN system has been created for the purpose of registering biopsy sites using MR images. The system generates noninvasive maps of cell density to identify the infiltrative margins of gliomas rather than just relying on the margins of the enhancing tumor alone [83]. Determining the level of surrounding infiltration can help to stratify patients into the appropriate invasive or noninvasive treatment options.

## 7. White Matter Disease

### 7.1. White Matter Hyperintensities

White matter lesions (WMLs), or white matter hyperintensities (WMHs), are areas of abnormal myelination that can be best appreciated as increased signal intensity on T2/FLAIR sequences. They are widely utilized neuroradiologic biomarkers of brain parenchymal pathologies. As a group, they include but are not limited to small-vessel ischemic disease, demyelinating disease, or other inflammatory processes [84,85]. The presence of these lesions has been shown to incur an increased risk of stroke, dementia, and death [86,87]. Furthermore, WMLs are associated with grey matter atrophy, accelerated neurodegeneration, and cerebrovascular incidents [88].

Due to the impact and location-dependent nature of these lesions, accurate detection and quantification is paramount. Traditional practices involve the manual delineation of regions via analysis dependent on visual, qualitative, or semiquantitative inspection of imaging, although even early studies recognized the benefit of automatic or semiautomatic image analysis. The utilization of neural networks and other AI approaches in quantifying white matter signal abnormalities and related findings such as brain volume changes has evolved over the past five years, with numerous methods already developed and commercially available [89,90,91,92,93,94,95,96]. Currently available software packages can include classification algorithms (e.g., diagnosis), regression algorithms (e.g., linking clinical scores or liquid biomarker levels to images), detection algorithms, segmentation algorithms, or a mix [97]. Several of the commercially available software packages are listed on the Grand Challenge website (https://grand-challenge.org/aiforradiology/, accessed on 15 May 2023).

Some of the most promising clinical applications of these algorithms have included use in the automatic and semiautomatic segmentation of multiple-sclerosis (MS) lesions, age-related WMH, and in aspects of radiomics [4,98]. All of these rely on the accurate quantification of lesions to determine prognosis, monitor treatment, and develop quantitative imaging biomarkers. Recent studies have found that automatic segmentation of these lesions has excellent accuracy and fares well compared to manual segmentation, with processing times ranging from 2–20 min. However, methods tend to underestimate the volume of lesions, and no current method factors in the wide variability of MRI contrast within subjects and protocols [98,99,100]. Furthermore, given the numerous available algorithms and packages, comparing options can be challenging. For this reason, comparison generally relies in part on the ECLAIR guidelines (Evaluation of Commercial AI Solutions in Radiology). However, issues remain on account of the variability of training datasets, patient populations, and many other factors. Perhaps most notably, segmentation of WMLs, despite being the most common primary task performed by AI, has not been well standardized for large-scale studies and has therefore historically struggled with automation [101,102]. Accurate quantification is a key area of research and exploration given that segmentation is a key factor in determining the extent of disease, the changes in clinical picture, and patient outcomes.

Similar to early public databases such as BrainWeb computational phantom, competitions like the International Conference on Medical Image Computing and Computer Assisted Intervention (MICCAI) and WMH Segmentation Challenge serve as catalysts for the development of novel methods and provide multi-institutional comparisons of algorithms using standardized evaluation criteria [103,104]. Several similar challenges exist that focus on different tasks, including multiple sclerosis (MS) lesions, tumors, and strokes [105,106,107,108]. Alhough the results of these varying competitions are largely incomparable due to differences in evaluation criteria and datasets, they provide valuable timelines for advancements in AI and ML in terms of radiological detection, segmentation, and quantification. Furthermore, some challenges such as the MS Lesion Segmentation Challenge occurred over multiple years (2015 and 2021) and were therefore able to provide insights into advancements within a specific application.

The 2017 MICCAI WMH Segmentation Challenge was held in Quebec, Canada, and originally included 20 participating teams. Teams were given a training dataset, containing 60 brain MR cases from three separate scanners, to develop their methods with and were then tested using a separate test dataset containing 110 MRI brain cases from five different scanners [103]. Though new methods are still being submitted, the winning method of the challenge demonstrated an F1 score of 0.76 and a Dice score of 0.80. This accuracy was achieved using an algorithm based on an ensemble of convolution–deconvolution architectures with 19-layer implementation optimized for classifying and localizing WMHs [25,102,109]. However, the algorithm did not perform well with multi-scale features, and in 2020 Liu and colleagues created a deep convolutional neural network, M2DCNN. This addressed the problem using two subnets that rely on a set of novel multi-scale features and a novel architecture designed to reduce the loss of receptive fields [110]. In addition to improved detection of large and small lesions, M2DCNN demonstrated fewer false positives and less variability in the predictive performance than previously described segmentation methods. Clinically, this advancement allows radiologists the ability to apply AI algorithms to images with minimal spatial information loss, improved image classification, and better localized segmentation [111]. Most recently, a new method built on the U-Net architecture achieved F1 scores as high as 0.93 using the MICCAI WMH challenge training dataset by introducing dense connections to allow for better utilization of multi-scale features [110].

### 7.2. Multiple Sclerosis (MS)

Multiple sclerosis is a debilitating disease that often affects people in the prime of their lives. The prevalence of MS is growing, and the need to have automated accurate detection and quantification of lesion burden is crucial to disease management and prognosis [112].

The first major MS lesion segmentation challenge occurred at the MICCAI 2008 conference. A second MS lesion segmentation challenge was conducted at the 2015 International Symposium on Biomedical Imaging (ISBI) in New York. The 2015 challenge and dataset are still being used and currently have over 2000 submissions. A subsequent 2016 segmentation challenge and dataset attempted to address multiple potential issues faced during the 2008 and 2015 challenges by using high-quality patient cases, drawn from four different scanners after delineation by seven expert evaluators, to reduce inter-rater variability. Evaluations utilized a distributed Web platform for the automatic, fair comparisons of algorithms. The challenge evaluated candidate algorithms’ performance for both detection (correct identification of all lesions in an image) and segmentation (precisely outline lesions). Only 13 teams participated in the original challenge in 2016. In this event, F1 lesion detection scores ranged from 0.13 to 0.49 (avg 0.32), with an expert range of 0.66 to 0.89 (avg 0.77). Dice segmentation scores for the algorithms ranged from 0.27 to 0.59 (avg 0.46), with the expert Dice scores ranging from 0.69 to 0.78 (avg 0.71) [108,113]. It was also noted that algorithm performance diminished with an increased number of lesions and a decreased size of lesions. The challenge was modified in 2021 to delineate new MS lesions with similar parameters and larger datasets. Although all experts still scored significantly higher than any submitted method, the difference was much smaller for detection, with combined average F1 scores of 0.61 for the experts compared to 0.42 for automatic methods [114]. Combined average Dice scores for the experts were 0.56 compared to 0.39 for the automatic methods. Detailed descriptions of each method and its results have been published in HAL Open Science, with comprehensive results available on the website (https://zenodo.org/record/5775523, accessed on 15 May 2023) [108,114].

## 8. Neurocognitive

Dementia is an acquired syndrome characterized by a significant decline in cognitive function that leads to difficulty in daily functioning/independence, with an estimated prevalence in the United States of 11% in individuals over 65 years old [115]. Mild cognitive impairment (MCI) represents cognitive impairment that is more severe than normal aging but which does not interfere with independent daily functioning. The screening, diagnosis, and monitoring of neurocognitive disorders are typically guided by a patient’s history and clinical symptoms, including an emphasis on the clinical interview [116]. Neuroimaging, specifically structural MRI, represents a widely available and non-invasive test when evaluating for cognitive dysfunction that is commonly used to support a diagnosis of cognitive impairment [117]. However, dementia and MCI are significantly underdiagnosed in the community setting, with up to 60% of cases not being detected [118]. The National Institute on Aging-Alzheimer’s Association Framework predicts that there also exists a silent preclinical stage of Alzheimer’s disease before cognitive symptoms emerge, with the possibility of novel interventions at this preclinical stage [119].

Because early and accurate diagnosis of dementia and other neurocognitive disorders is imperative to allow access to supportive therapies that can help patients maintain their independence, ML methods based on neuroimaging have great potential to promote earlier and more sensitive diagnosis of neurocognitive disorders [120]. For instance, ML algorithms utilizing electronic health record (EHR) data for early detection of cognitive impairment risk are being evaluated in an active clinical trial [121]. Because of the potential benefits of neuroimaging-based ML applications in the detection of dementia, there have been numerous studies investigating the applications of ML in this field.

Numerous challenges have been conducted to investigate the use of structural-MRI-based ML in the screening and diagnosis of dementia. While several challenges addressed research questions of aiding prediction of future outcomes and cognitive scores, the challenges that most pertain to the practicing neuroradiologist largely correspond to three clinical questions in dementia: screening, clinical status classification, and monitoring of disease progression [122,123,124]. The remainder of this section will briefly review current clinical practices in cognitive impairment screening, classification, and monitoring and discuss the Predictive Analytics Competition (PAC) 2019, Computer-aided diagnosis of Dementia (CADDementia) challenge, and Minimal Interval Resonance Imaging in Alzheimer’s Disease (MIRIAD) challenge, as well as their implications in their respective clinical tasks.

### 8.1. Screening

Currently, approaches to screening for cognitive impairment include screening tests such as the mini-mental state examination (MMSE), clock drawing test, and Montreal cognitive assessment (MoCA), among others, as well as biological markers [121]. However, in the most recent 2020 US Preventative Services Task Force (USPSTF) recommendation statement, they concluded that there existed insufficient evidence to assess the benefits and harms of widespread screening of asymptomatic adults for cognitive impairment [125]. Most primary care systems are not equipped to routinely detect dementia, especially in multicultural populations or those with lower levels of educational attainment [118]. These gaps in screening for cognitive impairment are an opportunity for neuroimaging-based AI algorithms to aid in clinical decision making.

The Predictive Analytics Competition (PAC) 2019 sought to improve ML models using whole-brain structural MRI scans from healthy individuals to predict brain age, a cumulative screening marker of functional capacity, residual lifespan, and the risk of progression to neurocognitive disease [126]. Participants were given the following tasks: 1. minimize the mean absolute error (MAE) between chronological and predicted age (brain-age gap) and 2. minimize brain–age gap while keeping the Spearman correlation between the brain–age gap and chronological age below *r =* 0.10 in order to reduce bias. Seventy-nine participating teams were provided with a large structural MRI training dataset of healthy individuals with ages provided (*N* = 2640) and tested on a test dataset obtained from the same institutions (*N* = 660). The winning team of Gong et al. used lightweight 3D convolutional neural networks (CNNs), combined with preprocessing steps and pre-trained on UK Biobank data (MAE = 2.95 years after bias correction) [127].

PAC 2019 revealed that current ML models are increasingly effective at predicting brain age without incurring significant bias. DL models also outperformed classic ML algorithms [126]. CNNs using 3D kernels instead of 2D kernels were useful in efforts to exploit features across all spatial dimensions. The top-performing models also utilized shallower CNN architectures in contrast to the deep 2D CNN architectures used in slice-level modeling [127], suggesting that either the sample size used in this challenge was too small for deeper structures to confer advantages, or that age-related brain morphology changes are relatively simple to detect [127].

While this challenge demonstrated the efficacy of ML models in the specific task of brain age prediction, the cost and availability of structural MRI acquisition for the general population of healthy adults as a screening test for cognitive impairment remains a barrier to the application of these models in clinical practice. Brain age prediction through these ML models may be more useful for populations that are predisposed to cognitive impairment, such as those with a family history or other known biomarkers.

### 8.2. Classification

In current clinical practice, the initial evaluation of cognitive impairment includes elements of clinical history, neurologic examinations with an emphasis on mental status, labs to screen for reversible causes of cognitive dysfunction (i.e., chemistries, thyroid panel, B12), and structural brain imaging, with MRI being the preferred method over CT [116]. Classification of cognitive impairment, which refers in this article to the distinction between cognitively normal patients (CN), those with MCI, and those with dementia, is primarily performed using elements of patient history and an assessment of cognitive function.

The CADDementia challenge was conducted in 2014 and gave participants the task of classifying baseline MRI scans into three diagnostic classes: Alzheimer’s disease, mild cognitive impairment (MCI) and cognitively normal (CN) [128]. Fifteen research teams submitted a total of 29 algorithms. The challenge provided a small training set of multicenter T1-weighted MRI scans (*N* = 30) of patients that were equally representative of the 3 diagnostic classes. Participants were also allowed to use extra training data, and most used data from the Alzheimer’s Disease Neuroimaging Initiative (ADNI) database and/or the Australian Imaging Biomarker and Lifestyle flagship study of aging (AIBL). Algorithms were tested against a previously unseen multi-center CADDementia dataset (*N* = 354) with clinical diagnosis established via a multi-disciplinary consensus that was blinded to the participants. The performance of the algorithms was quantified by classification accuracy, area under the receiver operating characteristic (ROC) curve (AUC), and the true positive fraction for the three classes.

The challenge submissions utilized a wide range of approaches, with most methods accounting for input features such as volume (most common feature, used by *N* = 19 algorithms), cortical thickness, intensity, and shape. Algorithms also utilized a variety of classifiers, such as SVM classifier, random forest classifier, and linear discriminant analysis (LDA). Only one algorithm (Folego-ADNet) used a convolutional neural network for classification and achieved a relatively low rank [129]. The best-performing algorithm used a LDA of features measuring volume, thickness, shape, and intensity relations of brain regions (accuracy = 63%, AUC = 78%) [130].

The results of the CADDementia challenge showed that the best-performing algorithms incorporated multiple features and utilized additional larger training datasets. While the performance of algorithms in this challenge was deemed too low for clinical application [122], multiple groups have applied DL models to ADNI data for this 3-class classification and achieved accuracies as high as 90% [131].

### 8.3. Assessing Disease Progression

In current clinical practice, monitoring for disease progression in dementia is largely centered on factors such as the loss of additional cognitive function, which may be quantified using the same screening instruments discussed earlier. Serial neuroimaging is not routinely performed in patients with dementia unless there is a new rapid loss of cognition, focal neurological signs, or seizure. There exist ongoing datasets of longitudinal biomarkers of dementia, such as the ADNI, although many of these biomarkers are obtained primarily in research settings and not part of standard clinical practice [132,133].

The Minimal Interval Resonance Imaging in Alzheimer’s Disease (MIRIAD) challenge was conducted to develop and compare methods of estimating atrophy and rates of atrophy from structural MRI [134]. In particular, the challenge looked at volumetric measurements of key structures: the whole brain, lateral ventricles, and hippocampus.

The MIRIAD dataset consisted of 708 T1-weighted volumetric scans acquired from 69 subjects (46 patients with clinical diagnosis of AD and 23 cognitively normal controls), with each subject undergoing 1–12 scans over 1–2 years at various time intervals. Submissions were graded based on the predicted sample size requirements for a hypothetical clinical trial, assuming a putative treatment effect of a 25% atrophy rate reduction [134]. The rationale for this approach was that the methodology with the most utility would require the smallest sample size to provide sufficient power, given that all other aspects of the design are fixed.

Challenge participants produced consistent and repeatable measures of change in brain regions and ventricles; however, hippocampal measures were the most variable among the submissions. Cash et al. attribute this variance to the differing definitions of the hippocampus used in segmentation protocols, as well as the possibility that the hippocampus is a small structure of the brain susceptible to MRI acquisition artifacts [134]. The best methods, i.e., those requiring the smallest sample sizes, were the boundary shift integral for whole-brain atrophy and the combination of diffeomorphic registration (Demons-LCC) and regional flux analysis for ventricle and hippocampus atrophy.

The application of the results of the MIRIAD challenge to clinical practice would be limited to the occasional cases of cognitive impairment in which serial neuroimaging is acquired. Several studies have attempted to predict the conversion of MCI into AD using baseline data, with accuracies as high as 83% when incorporating the results of multiple modalities, including structural MRI, PET, CSF, and clinical metrics [135]. This predictive task has the potential to permit the earlier identification of patients at highest risk for future progression to full-fledged dementia.

## 9. Spine

Back pain is a common problem and one of the leading causes of disability in both developed and developing countries [136,137]. Degenerative processes and spinal injuries also have increased in prevalence across all age groups, and the demand for imaging has followed suit [138,139]. MRI and CT are heavily relied upon to accurately diagnose spinal degenerative changes, deformities, instability, and fractures. This section will discuss the fundamentals of spine AI and current research on clinical applications.

Accurate spine labeling is a critically important aspect of imaging interpretation as it establishes the numerical and categorical relationship of one vertebra to another. Manual labeling can be time-consuming and error-prone, particularly when variant anatomy exists. In the setting of AI, several definitions pertain to the process of assigning a specific identifier to a vertebral body. Localization is the detection of a distinct three-dimensional unit in space (such as the vertebral body or the intervertebral disc), and labeling assigns an identifier or class to the three-dimensional unit. Segmentation can be thought of as a voxel-level labeling task [140]. Automated localization and segmentation will greatly increase accuracy and efficiency while also serving as a fundamental tool for additional AI-related spinal applications [140,141]. This includes diagnosis and evaluation of vertebral fractures, scoliosis, kyphosis, and degenerative intervertebral discs, which will be discussed later in this section.

Segmentation is a demanding task, both for radiologists and algorithms. While manual segmentation is possible, it is not practical in the clinical setting. For example, it is very time-consuming to segment the posterior elements manually. AI-based segmentation algorithms may suffer from an insufficient volume of adequate training datasets. In conjunction with the International Conference on Medical Image Computing and Computer Assisted Intervention (MICCAI), the Large Scale Vertebrae Segmentation Challenge (VerSe) was organized in 2019 and 2020 to tackle this ongoing problem. A large training dataset was created that included 374 spine CT exams annotated through semi-automated techniques, with additional manual refinement [140]. A total of 26 different algorithms were evaluated and compared based on the metrics of labeling accuracy and segmentation. In 2019, the best algorithms produced an identification (labeling) rate of 94.3% and a Dice score of 89.9%, which improved to 96.6% and 91.7% in 2020, respectively. Several specific impediments to accurate AI labeling and segmentation were fractures, metal implants, cement, and transitional vertebrae [140].

Intervertebral discs (IVD) have also been targets for automated localization and segmentation in efforts to help identify and quantify degenerative disc disease with the goals of minimizing error and time spent on manual interpretation [141,142]. Li et al. created an automated method that received first place in the MICCAI 2016 automatic IVD challenge, with a mean segmentation Dice coefficient of 91.2% and a mean localization error of 0.62 mm on multi-modal magnetic resonance images (MRI) [141]. The low performance of the segmentation algorithms primarily involved disc margins, which commonly can have variable and irregular appearances, which the authors suggested could be improved with higher image resolution [143]. As with variation in vertebral anatomy, variations in disc morphology may also affect the segmentation of discs.

There are numerous applications once accurate segmentation is achieved, including the evaluation of degenerative changes. In 2017, Jamaluden et al. developed a localization algorithm using T2-weighted sagittal spine MRI images with an accuracy of 95.6% [144]. Their algorithm was also able to grade canal stenosis, Modic type changes, and disc narrowing, as well as perform Pfirrmann grading at a level comparable to a radiologist [144,145]. Hallinan et al. developed a DL model using a set of 446 MRI lumbar spine studies (T2W axial and T1W sagittal) that classified central spinal canal, lateral recess, and foraminal stenosis using a dichotomous scale (normal/mild vs. moderate/severe [146]), with performance comparable to 2 radiologists [146]. The DL algorithm was slightly worse in terms of agreement when tasked with assigning four distinct levels of severity (normal, mild, moderate, severe).

Vertebral fractures in the thoracic and lumbar spine have also been targets of AI automation and assistance. Compression fractures may be overlooked by radiologists, especially those who do not frequently read spine imaging [147]. Burns et al. reported an algorithm with the ability to detect and localize thoracic and lumbar spinal fractures on CT scans from 150 patients with a sensitivity of 95.7% and a false positive rate of 0.29 per patient [147,148]. The authors were also able to classify by Gerant type (anterior, middle, and posterior height loss) with an accuracy of 95%. Murata et al. trained a DL model with AP and lateral thoracolumbar radiographs on 300 patients that could localize fractures with the accuracy and sensitivity of 86% and 84.7%, respectively [149]. They demonstrated their model’s ability to detect vertebral fractures to be equivalent to that of orthopedic surgeons and residents [149,150]. In conjunction with the American Society of Neuroradiology (ASNR) and American Society of Spine Radiology (ASSR), the Radiological Society of North America (RNSA) created a challenge in 2022 to encourage the creation of AI-based algorithms to detect and localize cervical spinal fractures. Their dataset included around 3000 normal and fracture-positive CT examinations annotated by expert spinal radiologists from ASNR and ASSR.

The evaluation of scoliosis is another prime target for AI, including Cobb angle measurements. Issues of reproducibility, accuracy, and the inter-rater reliability of clinicians manually measuring these angles could potentially be addressed using AI. Wang et al. proposed a multi-view extrapolation net (MVE-Net) to estimate Cobb angles using AP and lateral radiographs [151]. The authors obtained a circular mean absolute error of 7.81 degrees on AP and 6.26 degrees on lateral x-ray angle estimation on 526 images, presenting a reasonably accurate estimation of the degree of scoliosis. Large-scale competitions, such as the Accurate Automated Spinal Curvature Estimate (AASCE2019) challenge, have demonstrated strong performances among many algorithms [152]. The AASCE2019 challenge employed a dataset of 707 spine AP radiographs and evaluated algorithms’ performance at producing accurate and reliable Cobb angles. Zhang et al. also used an artificial neural network to measure Cobb angles, examining 65 in vivo coronal radiographs (patients with idiopathic scoliosis) and 40 model radiographs (from a spine model positioned in different poses) [153]. Their model had an absolute error of less than 3 degrees for the spine model radiographs but performed worse on the in vivo images.

## 10. Head and Neck

### 10.1. Tumors

Artificial intelligence (AI) has the potential to revolutionize head and neck imaging by augmenting image quality and improving its ability to perform clinically relevant tasks such as tumor volume segmentation, tumor characterization, tumor prognostication, treatment response assessment, and the prediction of metastatic lymph node disease [154,155]. Head and neck oncology care is well positioned for the application of imaging AI since treatment is guided by a wealth of information derived from US, CT, and MRI imaging data. DeJohn et al. conducted a literature review of the current state of the field and identified several areas where ML could potentially be applied, including image quality enhancement, automatic feature extraction, and automated diagnosis [156]. ML and DL models can improve patient care throughout the clinical workflow from the time of imaging to interpretation and through quality improvement via standardization of automated tools.

AI has also been applied in the prognostication of responses to chemotherapy or radiation in head and neck cancer [157]. In addition to auto-segmentation for treatment planning, AI tools can also be beneficial in oncological outcome prediction and toxicity prediction in radiation treatment [158]. The Head and Neck Organ-at-Risk Multi-Modal Segmentation Challenge (https://han-seg2023.grand-challenge.org/, accessed on 15 May 2023) was launched recently to promote the development of new and existing applications of fully automated techniques for OAR (organ-at-risk) segmentation in the head and neck regions of CT images. The goal of this challenge is to exploit the information of multiple imaging modalities in order to improve the accuracy of segmentation results. The HEad and neCK TumOR (HECKTOR) challenges (https://hecktor.grand-challenge.org/, accessed on 15 May 2023) focused on establishing best-performing methods in order to predict patient outcomes from FDG-PET/CT and clinical data and conduct the automatic segmentation of head and neck primary tumors and lymph nodes on FDG-PET/CT images.

AI has the potential to significantly improve the accuracy and efficiency of ultrasound use in head and neck oncology. A systematic review by Santer et al. found that 74% of studies on the use of AI in ultrasound for head and neck oncology addressed disease diagnosis, with 56% examining the ability to distinguish benign and malignant thyroid nodules and 44% seeking to identify metastatic lymph nodes [157]. Radiomics-based MRI features have been used in the assessment of various head and neck cancer (HNC) lesions. In several studies, traditional ML techniques have been used for the automatic segmentation of HNC lesions using MRI, with promising results in terms of accuracy (86 ± 8%) and overlap measures (0.76+/−0.08) [159,160]. The textural analysis of MRI and CT images has also been used to differentiate between different types of HNC lesions, with accuracies ranging from 75.7% to 100% [161,162,163]. Parameters obtained via histogram and texture analysis of MRI T2WI can even serve as noninvasive predictors of histological type and grade in head and neck malignancy [164]. In addition, textural features derived from intraoral X-ray images have been used to predict the early onset of oral squamous cell carcinoma, with accuracies of 99.2% [165]. Overall, these findings suggest that radiomics-based prediction can be a useful tool for the assessment and diagnosis of HNC lesions.

The increasing number of independent prognostic and predictive markers has sparked interest in the use of AI-based prediction models. AI-based methods can integrate complex imaging, histologic, molecular, and clinical data to model tumor biology and behavior, and potentially identify associations far beyond what conventional qualitative imaging can provide alone. DL-based models can more accurately predict oncological outcomes using pre-treatment data than existing models. Likewise, they are better at predicting treatment toxicity prior to the start of treatment as well as the prediction of pathological data from imaging data [166].

### 10.2. Vascular Lesions

CTA is a widely used and cost-effective imaging modality for the diagnosis of cerebrovascular disease in the head and neck region. However, manual postprocessing of CTA images can be time-consuming and subject to human error. DL-based segmentation approaches have been proposed as having the potential to improve the efficiency of CTA analysis by reducing the need for manual post-processing. One major challenge in CTA image postprocessing is the accurate segmentation of vessels, exacerbated by their branching morphology, variable anatomy, and overlap in density with other tissues. To address these challenges, Fu et al. developed an automatic imaging reconstruction system called CerebralDoc, which uses a 3D-CNN containing modified U-net components for the reconstruction of original head and neck CTA images [167]. This system has the potential to assist CT technologist or radiologist workflow and improve efficiency by removing time-consuming steps in CTA post-processing.

Traditionally, the noninvasive diagnosis of cerebral aneurysms has relied on imaging modalities such as CTA or magnetic resonance angiography (MRA). However, these techniques can be limited in their ability to accurately detect and classify small or complex aneurysms. AI has the potential to improve the accuracy of cerebral aneurysm diagnosis by automating the analysis of imaging data and identifying subtle features that may be overlooked by human observers. In a study by Park et al., a DL-based model called HeadXNet was developed for the diagnosis of cerebral aneurysms using CTA images [168]. The model alone had a sensitivity of 0.95 and specificity of 0.66 for the detection of aneurysms. When paired with trained radiologists, the model improved a radiologist’s sensitivity, accuracy, and interrater agreement. Chen et al. developed a DL-based method for the segmentation of cerebral aneurysms in 3D TOF-MRA images using a coarse-to-fine framework [155]. The method was able to accurately identify and segment aneurysms, with a Dice coefficient of 0.87.

DL-based models have also been developed in order to improve the accuracy of cerebral aneurysm rupture risk prediction. In a study by Yang et al., a CNN-based DL model was developed for the prediction of cerebral aneurysm rupture risk using 3D time-of-flight magnetic resonance angiography (TOF-MRA) images [169]. The model was able to achieve high accuracy in predicting rupture risk, with an AUC of 0.95 [169]. Similarly, AI has the potential to improve the accuracy and efficiency of venous malformation diagnosis and treatment planning. In a study by Ryu et al. (2022), a DL-based method called 3D U-Net was used for the automatic segmentation of extracranial venous malformations in the head and neck region from MRI images [170]. The method was able to accurately identify and segment venous malformations with a high degree of accuracy, producing a Dice coefficient of 0.87.

## 11. Conclusions

Artificial intelligence has numerous applications throughout the field of neuroradiology, with great promise to augment the work of the modern-day radiologist. AI-based methods have taken large strides in accuracy and efficiency over the past decade, some of which can be linked to the AI challenge competitions conducted to benchmark leading AI methodologies using well-annotated datasets. As demonstrated in this article, AI applications have been developed and evaluated for use in detecting or quantifying intracranial hemorrhage and stroke, brain and head/neck tumors, spinal fractures, degenerative spinal disease, and inflammatory or neurodegenerative brain disorders. While many AI methods demonstrate remarkable performance in specific tasks and several software packages have been approved for clinical use, there remains a continuous need to push the field further to develop improved tools with which to better augment the work of the practicing neuroradiologist. AI-based challenges will likely continue to showcase the latest advancements in AI methods and provide an impetus for improvement, ultimately leading to higher-quality patient care.

## Figures and Tables

**Figure 1 diagnostics-13-02670-f001:**
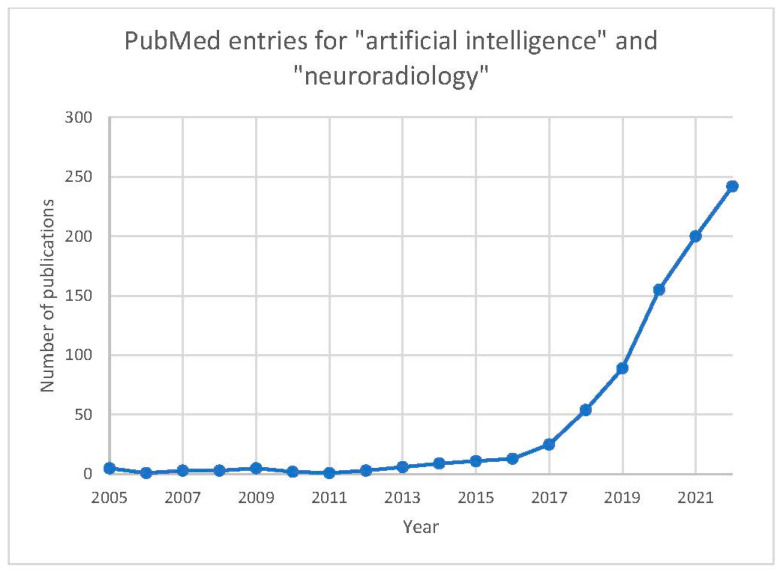
Counts of PubMed entries obtained when searching for “artificial intelligence” and “neuroradiology” by calendar year of publication.

**Figure 2 diagnostics-13-02670-f002:**
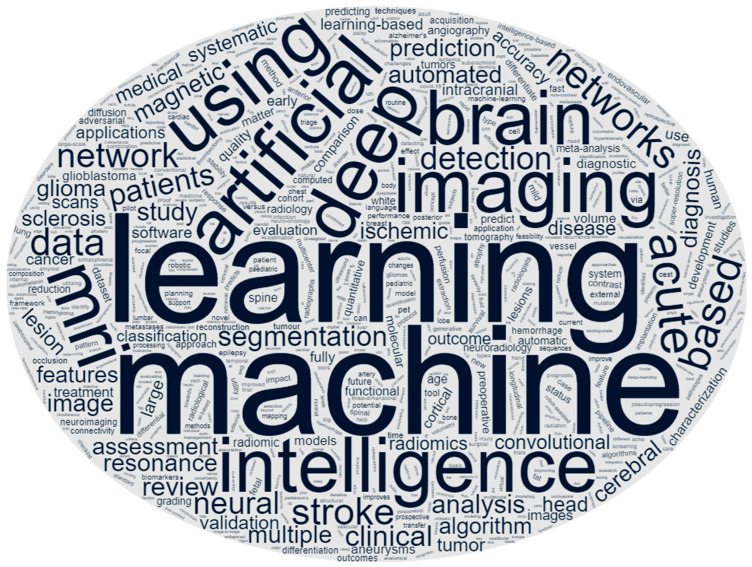
Word cloud depiction of terms from titles of articles from a PubMed query for “artificial intelligence” and “neuroradiology” published in 2017 or later.

## Data Availability

Not applicable.

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
