# Peer review of "Artificial Intelligence in Neuroradiology: A Review of Current Topics and Competition Challenges"

_diagnostics, 2023, doi:10.3390/diagnostics13162670_

Round 1
Reviewer 1 Report
Manuscript ID: diagnostics-2426589
Title: Artificial intelligence in neuroradiology: a review of current topics and competition challenges.
Comments and remarks:
This paper proposes a literature review of artificial intelligence in neuroradiology including a wide range of applications. The authors aimed at tackling localization, segmentation, longitudinal monitoring, disease classification, and prognostic objectives.
The paper is of good quality and well-structured. Some remarks and comments can be considered to improve the readability of the review.
1. Figure 1 should include legends in x and y axes.
2. Figure 2 can be regenerated in better resolution and more clear words.
3. Authors mentioned, “AI challenge competitions are events organized around a specific task or problem. ». Details on the specific tasks can be reported for further positioning for future researchers.
4. In the AI challenge Competitions section, it is mentioned that AI-based attempts to provide annotated images. More details on what is annotation and different annotation types can be reviewed too.
5. “Deep learning (DL) is a subtype of ML that uses multi-layer artificial neural networks”. In my opinion, we can use a long-short-term memory (LSTM) model, which is a deep neural network, but without a lot of layers. In this case, LSTM is also considered a deep neural network. Hence, the definition should be revised properly.
6. Evaluation metrics should be described by equations also.
7. “One major obstacle to robust performance, however, lies in the single-center design and the small size of single-institution training and test datasets”. Here, the lack of datasets is presented as a unique main obstacle for AI. I think that data quality is a key issue for AI which is not addressed in this paper.
8. Detection diagnostic and prognostic terms should be defined in the paper because definitions can be confused with the same in other domains.
9. In all the paper reviews, the focus was made on the AI models only. The authors do not specify if the reported works used the datasets in the raw form or were preprocessed, and processed before. This is crucial information to open new research directories. It should be summarized or pointed out in a new section with discussion.
Technical English needs to be revised for appropriate terms use.
Author Response
We thank the reviewers for providing helpful feedback on our manuscript. We have revised the manuscript, which we believe has improved in quality and readability. The manuscript has been reviewed by authors fluent in English scientific writing. Responses to individual comments are listed below in bold-type.
This paper proposes a literature review of artificial intelligence in neuroradiology including a wide range of applications. The authors aimed at tackling localization, segmentation, longitudinal monitoring, disease classification, and prognostic objectives.
The paper is of good quality and well-structured. Some remarks and comments can be considered to improve the readability of the review.
- Figure 1 should include legends in x and y axes.
Axis labels have been added.
- Figure 2 can be regenerated in better resolution and more clear words.
This figure has been re-rendered to improve resolution of the image. Please keep in mind that the purpose of the word cloud is to illustrate the most common terms encountered, and we do not intend for all of the text (particularly the low-frequency terms depicted in fine print) to be readable.
- Authors mentioned, “AI challenge competitions are events organized around a specific task or problem. ». Details on the specific tasks can be reported for further positioning for future researchers.
Details of the specific tasks are provided in the respective subsections of the text. In addition, further details regarding the overall design of these AI competitions have been added to the AI Challenge Competitions paragraph to help describe the typical tasks to be performed by candidate algorithms.
- In the AI challenge Competitions section, it is mentioned that AI-based attempts to provide annotated images. More details on what is annotation and different annotation types can be reviewed too.
Examples of annotations have been added to the 2nd paragraph of the AI Challenge Competitions section.
- “Deep learning (DL) is a subtype of ML that uses multi-layer artificial neural networks”. In my opinion, we can use a long-short-term memory (LSTM) model, which is a deep neural network, but without a lot of layers. In this case, LSTM is also considered a deep neural network. Hence, the definition should be revised properly.
We believe the definition of deep learning we provided accurately reflects its usage in the AI literature. A basic LSTM containing a single hidden layer would not be considered a deep network. While deep learning models may use LSTM, this is accomplished by stacking multiple LSTM units to achieve the desired depth.
- Evaluation metrics should be described by equations also.
Mathematical expressions have been added to define each of the evaluation metrics.
- “One major obstacle to robust performance, however, lies in the single-center design and the small size of single-institution training and test datasets”. Here, the lack of datasets is presented as a unique main obstacle for AI. I think that data quality is a key issue for AI which is not addressed in this paper.
In paragraph 2 of the AI Challenge Competitions section, we had included the following statement: “A critical problem with AI in radiology is the lack of large, public, high-quality, well-annotated data sets.” In addition, we added another statement to this paragraph to emphasize the data quality issue: “In some situations, large amounts of data are available but may not have been curated to ensure high data quality or adequate data standardization.”
- Detection diagnostic and prognostic terms should be defined in the paper because definitions can be confused with the same in other domains.
Definitions of these terms have been added to this paragraph.
- In all the paper reviews, the focus was made on the AI models only. The authors do not specify if the reported works used the datasets in the raw form or were preprocessed, and processed before. This is crucial information to open new research directories. It should be summarized or pointed out in a new section with discussion.
A few sentences have been appended to the first paragraph of the AI Challenge Competitions section to clarify the nature of the datasets and what processing is typically performed on the images.
Reviewer 2 Report
The manuscript offers a comprehensive literature review on the application of AI in neuroradiology, with topics ranging from the utilization of large datasets to algorithm evaluations and a diverse array of applications for different medical conditions. However, the present analysis lacks in-depth insights and recommendations for future research endeavors. Overall, the current manuscript demonstrates a commendable effort in providing a broad and informative perspective on the topic of interest. However, a more comprehensive analysis is necessary in promoting further investigations and progress in this field.
Major concerns:
1. As a comprehensive review article, it is imperative to provide an adequate description of the literature selection process. However, the present manuscript falls short in providing a necessary overview of the literature screening process, thereby posing significant limitations in evaluating the reliability and validity of the final literature pool.
2.The article provides extensive descriptions of different application fields and their corresponding competitions, but lacks necessary overall discussion and summaries. As a result, the content appears scattered and limited in value for readers. In fact, many review papers provides in depth discussion of AI methods for each subfield that author discussed. It is suggested that that author can combined these with his discussion.
3. The intended audience of this article may possess limited understanding of AI competitions and AI algorithms. Therefore, it is recommended for the author to include a chapter that briefly describes the fundamental design of competitions and provides a concise overview of the AI algorithms extensively discussed in the text.
Author Response
We thank the reviewers for providing helpful feedback on our manuscript. We have revised the manuscript, which we believe has improved in quality and readability. The manuscript has been reviewed by authors fluent in English scientific writing. Responses to individual comments are listed below in bold-type.
The manuscript offers a comprehensive literature review on the application of AI in neuroradiology, with topics ranging from the utilization of large datasets to algorithm evaluations and a diverse array of applications for different medical conditions. However, the present analysis lacks in-depth insights and recommendations for future research endeavors. Overall, the current manuscript demonstrates a commendable effort in providing a broad and informative perspective on the topic of interest. However, a more comprehensive analysis is necessary in promoting further investigations and progress in this field.
Major concerns:
- As a comprehensive review article, it is imperative to provide an adequate description of the literature selection process. However, the present manuscript falls short in providing a necessary overview of the literature screening process, thereby posing significant limitations in evaluating the reliability and validity of the final literature pool.
The review article was not intended to be a meta-analysis or systematic review. Similar to most review articles published in this journal, we emphasized literature published in the last 5 years, but we did not implement rigorous screening or evaluation methodologies that are typically reserved for meta-analyses. Instead, we made subjective determinations based on our clinical and research experiences to produce a review that sufficiently covers “tasks central to neuroradiologists,” as we stated in the introduction. To further clarify the literature selection process, we have also now added to the introduction an additional phrase to describe our emphasis on literature published in 2017 or later.
2.The article provides extensive descriptions of different application fields and their corresponding competitions, but lacks necessary overall discussion and summaries. As a result, the content appears scattered and limited in value for readers. In fact, many review papers provides in depth discussion of AI methods for each subfield that author discussed. It is suggested that that author can combined these with his discussion.
A short summary sentence has been added for some of the major subtopics. In addition, the conclusion paragraph has been revised to summarize the major applications discussed.
While many reviews are available to discuss the wide variety of AI methods, our current manuscript is focused primarily on summarizing the clinical neuroradiology applications of available AI tools rather than the technical methodologies underlying them. While such in-depth discussion of AI methods could add value, we believe there is sufficient existing literature covering that topic, and inclusion of in-depth discussion of AI methods may unnecessarily lengthen this article and detract from the intended focus of the manuscript on clinical neuroradiology applications.
- The intended audience of this article may possess limited understanding of AI competitions and AI algorithms. Therefore, it is recommended for the author to include a chapter that briefly describes the fundamental design of competitions and provides a concise overview of the AI algorithms extensively discussed in the text.
Additional statements have been added to the first paragraph of the AI Challenge Competitions paragraph to provide a brief overview of the design of AI challenge competitions. Additional text has been added to the Definitions section describing an overview of common AI algorithms discussed in the text; in addition to the previously included statement introducing CNNs in DL architectures, a brief statement introducing U-net architectures has now been added.
More detailed descriptions of AI algorithms are provided in the designated sections for each subtopic, along with cited references.